# S.M.A.R.T. F.U.S: Surrogate Model of Attenuation and Refraction in Transcranial Focused Ultrasound

**Joshua A. Cain**[1]*, **Shakthi Visagan**[2], **Martin M. Monti**[1,3]

**1** Department of Psychology, University of California Los Angeles, Los Angeles, California, United States of America, **2** Department of Neurology, University of California Los Angeles, Los Angeles, California, United States of America, **3** Department of Neurosurgery, University of California Los Angeles, Los Angeles, California, United States of America

* joshcain@ucla.edu

**Data Availability Statement:** held or will be held in a public repository https://doi.org/10.5281/zenodo.5811122 doi:10.5068/D1QD60.

**Funding:** National Institutes of Health, R01-NS074980, SV. The funders had no role in study

## Abstract

Low-intensity focused ultrasound (LIFU) is an increasingly applied method for achieving non-invasive brain stimulation. However, transmission of ultrasound through the human skull can substantially affect focal point characteristics of LIFU, including dramatic attenuation in intensity and refraction of focal point location. These effects depend on a high-dimensional parameter space, making these effects difficult to estimate from previous work. Instead, focal point properties of LIFU experiments are often estimated using numerical simulation of LIFU sonication through skull. However, this procedure presents many entry barriers to even computationally savvy investigators and often requires expensive computational hardware, impeding LIFU research. We present a novel MATLAB toolbox (data: doi:10.5068/D1QD60; Matlab Scripts: https://doi.org/10.5281/zenodo.5811122) for rapidly estimating beam properties of LIFU transmitted through bone. Users provide specific values for frequency of LIFU, bone thickness, angle at which LIFU is applied, depth of the LIFU focal point, and diameter of the transducer used and receive an estimation of the degree of refraction/attenuation expected for the given parameters.

## Introduction

In recent years, low-intensity focused ultrasound (LIFU) has gained popularity as a novel neuromodulation method due to its noninvasive implementation, relative ease-of-use, and theoretically highly precise targeting of both cortical and sub-cortical brain regions [1]. A major challenge in the implementation of LIFU is the degree to which skull absorbs, reflects, and diffracts ultrasound [2]. Skull dramatically affects the intensity of LIFU reaching the brain, often attenuating it by more than 50% in humans [3, 4]. Skull also affects the location of LIFU foci by refracting incoming sound waves (a total translation of 1cm is not uncommon [5]). Thus, the realization of LIFU as a safe and highly precise form of neuromodulation depends on the proper estimation of these qualities. However, the relationship between relevant LIFU

design, data collection and analysis, decision to publish, or preparation of the manuscript.

**Competing interests:** The authors have declared that no competing interests exist.

parameters and the energy pattern realized inside the skull have been relatively underexplored compared to other noninvasive techniques (e.g., TMS).

While pioneering studies provide a general understanding of the relationship between some of these parameters and skull propagation [5, 6], their results may not be readily used to estimate effects of skull in the context of any singular experimental preparation, which likely differs from previous investigations in at least one parameter. Although many studies employ simulations to estimate energy deposition inside the brain given specific parameters and experimental settings [7], this procedure is often impractical due to the computational power and sophistication required.

We introduce the Surrogate Model of Attenuation and Refraction in Transcranial Focused Ultrasound (SMART FUS) toolkit, which provides users the ability to estimate the effects of skull without intensive simulations of acoustic propagation. We achieve this using precomputed simulations that sample a wide range of the LIFU parameter space (a surrogate model). We distribute this dataset with a user-friendly MATLAB program (https://doi.org/10.5281/zenodo.5811122) that enables users to input study-specific experimental parameters and rapidly receive: **1)** estimated attenuation in focal intensity inside the brain; **2)** estimation of the degree of translation of the focal point; and **3)** visualizations of the precomputed simulations nearest to the provided parameters. SMART FUS aims to greatly streamline the process of planning LIFU parameter sets and trajectories that are reasonable given the often dramatic impact of the human skull.

## Usage

SMART_FUS includes two functions: SMART_FUS and SMART_FUS_vis2d.

SMART_FUS takes as input values for each dimension studied and returns an estimate of attenuation and refraction based on linear interpolation of two datasets that sample the attenuation and refraction in the parameter space (see below). Additionally, the nearest simulation point is visualized, providing the user an estimate of focal shape for the given parameters (see Fig 1A).

SMART_FUS_vis2d takes as input values for any three of the five dimensions studied and returns two 2D images depicting the refraction and attenuation expected at points across the two unspecified dimensions, which represent 2D slices through the full 5D parameter space. These 2D arrays are extracted and linearly interpolated to upscale the output matrix and provide a smooth space for visualization (see Fig 1B).

## Materials and methods

No IRB review was performed for this work as no participants were necessary. We defined a five-dimensional parameter space, comprising carrier frequency, bone thickness, trajectory, transducer diameter, and focal depth (detailed in Fig 2A), in which we simulated acoustic waves through bone. In this case, trajectory refers to the angle at which the transducer meets the bone surface. Some portions of this parameter space placed the theoretical focus either inside bone or outside the head entirely; inputting these values into SMART_FUS will return an error. Similarly, inputting values outside the parameter space studied will also return an error. In all, 12,096 simulations were run using an NVIDIA RTX 3080 GPU with each simulation taking roughly 10 seconds to complete, making for approximately fourteen days of simulation time.

We applied the MATLAB-based pseudospectral solver toolbox k-Wave [7] to perform the simulations(see Aubry et al., 2022 [8] for comparison to other comparable software). Note that K-wave itself has been extensively validated empirically [9–12]. A simulation space of

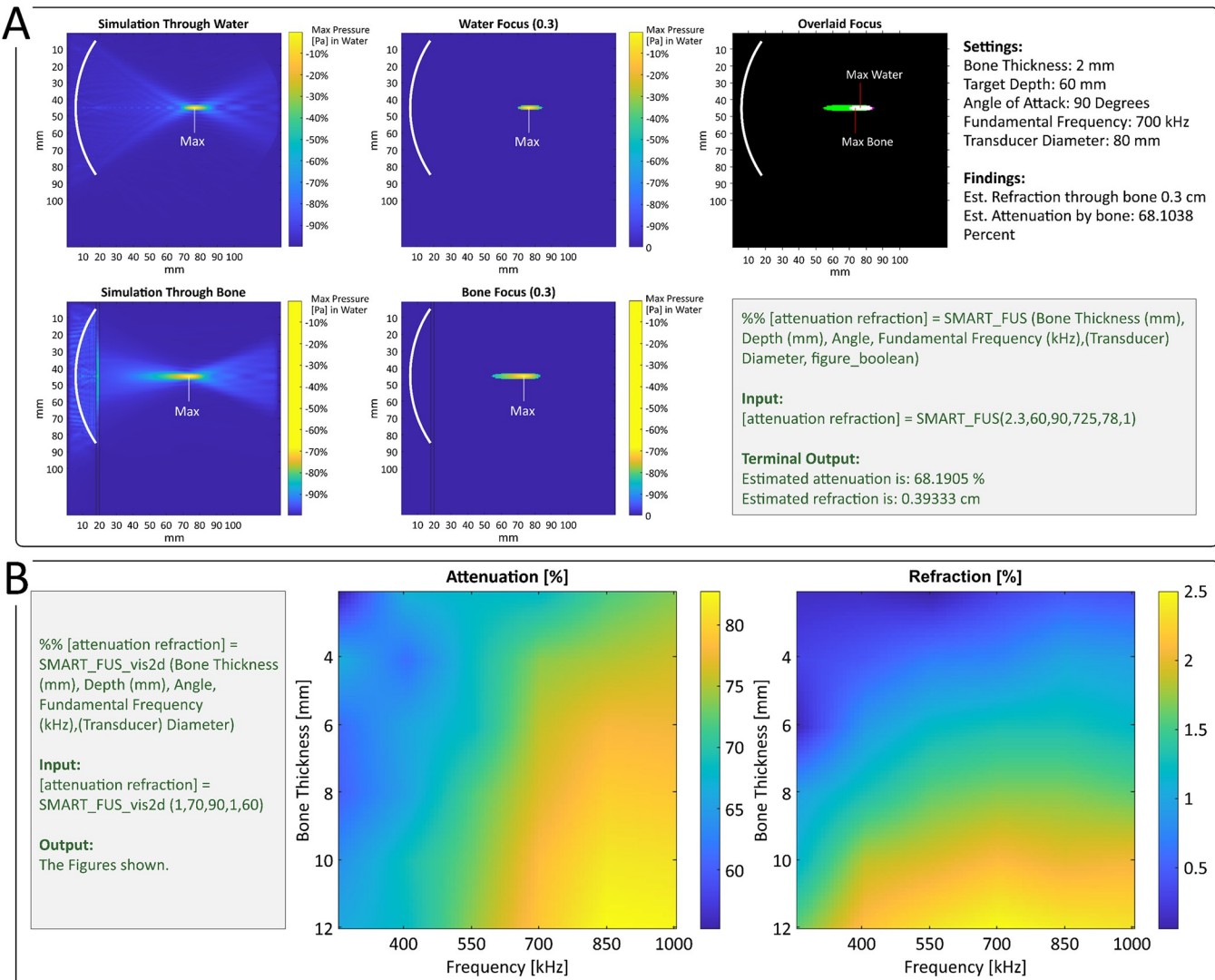

**Fig 1. Primary graphical outputs from SMART_FUS. A)** SMART_FUS estimates attenuation and refraction of specific inputs through linear interpolation and also provides a depiction of the simulation with parameters nearest the inputs. Here, a simulation conducted with a bone thickness of 2mm, target depth of 60mm, angle of entry of 90 degrees, a fundamental frequency of 700kHz, and a transducer diameter of 80mm is shown both when performed through a medium containing bone and when performed through a medium containing only water. The -3dB focal region (equivalent to the region exposed to 50% or more of the maximum intensity) is depicted to the right of the full simulations. The point of maximum intensity is highlighted. Finally, against a black background, these -3dB focal regions are overlaid and their points of maximum intensity highlighted again to clearly demonstrate the impact of bone on focal properties. It is important to note that this output would be generated if a close but not identical parameter set were inputted (e.g., with a bone thickness of 2.5mm and all other parameters remaining constant). However, the estimated attenuation and refraction with this unique parameter set would be generated through interpolation and output in the MATLAB terminal. **B)** A sample output from the script SMARTF_FUS_vis2d.m. Here, the user selected to visualize the impact of bone thickness and carrier-wave frequency at specific values for the remaining three free parameters. This 2-dimensional space is shown for both attenuation and refraction.

$256 \times 256 \times 256$ voxels with isotropic dimensions of 0.5mm each was created. Using dimensions that were powers of two maximized the speed of the Fast Fourier Transform used in k-Wave's Fourier collocation method. A perfectly matched layer (PML) was created to absorb energy at the edge of the simulation; default values of 10 voxels and a $PML_{alpha}$ of 2 were used. A Courant-Friedrichs-Lewy (CFL) of 0.3 (the default value for k-Wave) was chosen. A trade-off between runtime and simulation fidelity exists for CFL values. A value of 0.3 is appropriate for simple simulations, such as these, and maintains feasible simulation times. When ignoring

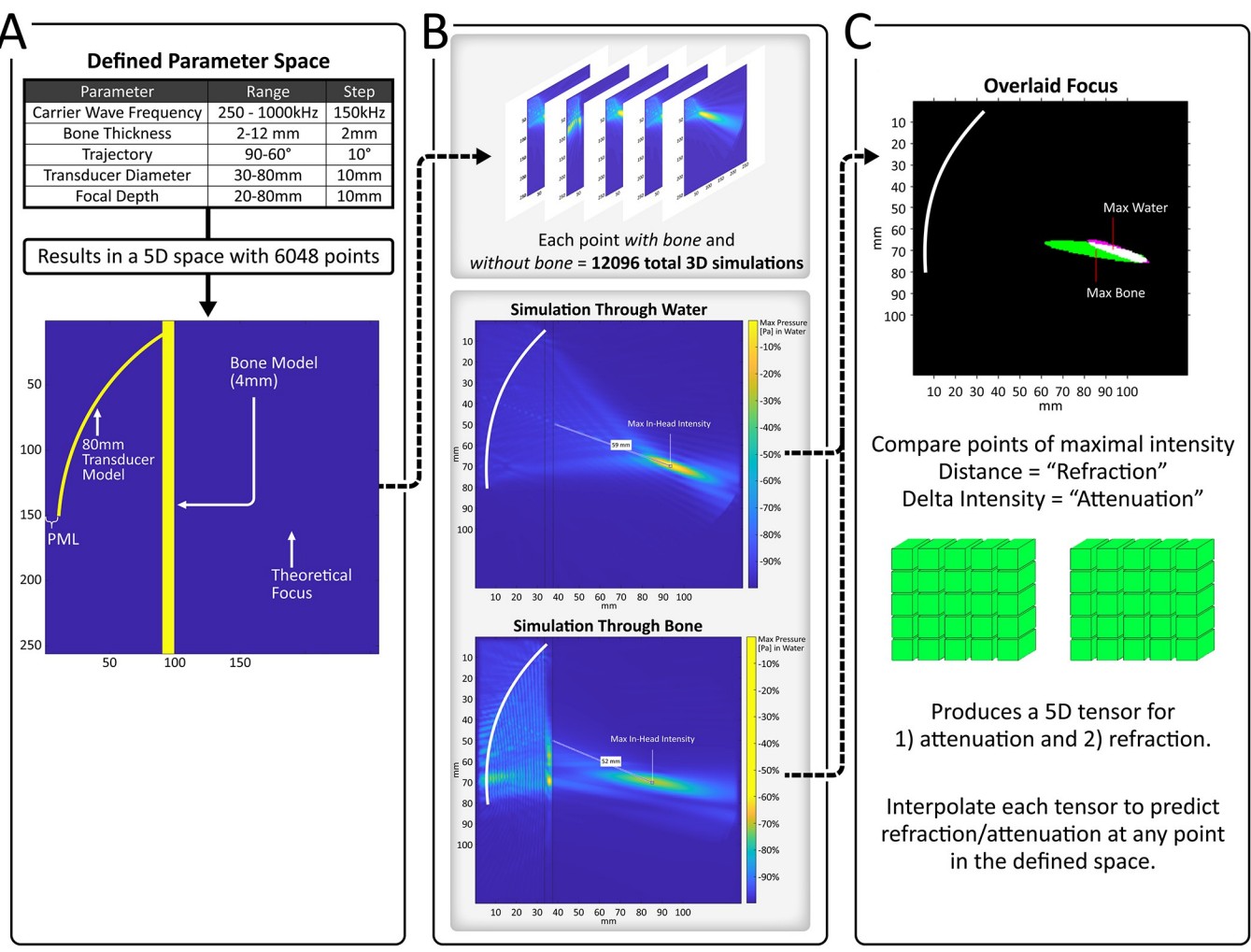

**Fig 2. Workflow for the creation of the dataset provided in SMART_FUS.** As described in more detail in the Methods section, a 5-dimensional parameter space was first defined and discretized. From this, 6,048 unique simulation environments were constructed (shown in **A**). Simulations with bone and water as well as through only water were run for each of these environments (resulting in 12,096 total simulations). The value and position of the peak intensity were located for bone and water simulations at each of the 6,048 points in the space defined. The distance between these points defined the observed "refraction" and the difference in intensity between these points defined the observed "attenuation" at each point in the space. Thus, one 5D "refraction" tensor (an array of 3D matrices) and one 5D "attenuation" tensor were created. These tensors are linearly interpolated according to the values input into SMART_FUS and are used to predict attenuation/refraction at any point within the described parameter space.

the PML layer, our simulations represent a space of 11.8 cm in each direction, capable of subsuming the necessary components of each simulation: a transducer up to 80mm in diameter with a focal depth up to 80mm, and a bone thickness up to 12mm. These dimensions, given the frequencies sampled, result in a points-per-wavelength (PPW) ranging from 11.86 (at the lowest frequency) to 2.96 (at the highest frequency). Homogenous 3D simulations have been observed to converge (in terms of recorded intensity and focal location) at a PPW of 3.5, even with more complex skull models [11] with values lower than this introducing some error. As our medium contains non-aliased skull-models (discussed more below) and a lower CFL of 0.3 (resulting in higher fidelity), compared to the cited study's CFL of 0.5, a slightly lower PPW of 2.96 (compared to 3.5) when simulating with a frequency of 1MHz remains valid. The PPW is higher than 3.5 when lower frequencies are used, but see the Validation section below for convergence testing and other validations.

For each simulation, a single-element transducer was modeled and assigned as the source of the acoustic waveform. The modeled transducer has dimensions that corresponded to a certain radius of curvature and diameter. The radius of curvature was determined using the desired focal depth of the transducer with the simple formula:

$$Radius\ of\ Curvature = \sqrt{(Transducer\ Focal\ Depth)^2 + (Transducer\ Radius)^2}$$

Transducers were generated in voxel space using k-Wave's makeBowl function and placed such that they met the upper-left corner of the PML. Bone was assigned to a flat wall of voxels in front of and meeting the transducer face. While transducers assigned to a perpendicular trajectory met bone along its entire face, those assigned a non-perpendicular trajectory were tilted away from the PML so their foci fully avoid the perimeter of the space. Regardless of trajectory, the voxel structure of bone remained flat to avoid an aliased surface (see Fig 1). Since aliasing has been characterized as the most significant source of error in simulations of this type [11], this likely enhances the fidelity of our simulations considerably.

Each voxel in the space not corresponding to bone was assigned a density of 1000 kg/m$^3$ and speed of sound of 1482 m/s, corresponding to that of water. A simulation time of $8.0375e^{-5}s$ was chosen as this allowed the simulated acoustic wave to reach the opposing end of the simulation space. The intensity emitted from the transducer was set to 1.0558MPa. Because all outputs from this function reflect the relative attenuation/refraction between water and bone simulations, this value is arbitrary but was selected from measurements taken from a previous empirical study [3] (see **Discussion**). Because the absorption coefficient for water changes in relation to the frequency of sound being simulated, we calculated it using k-Wave's waterAbsorption function by inputting the frequency used and a temperature of 37c (body temperature). Bone attenuation was set to $85 Np\ Mhz^{-1}m^{-1}$, while speed of sound in bone was set to 2850m/s, and Bone density was set to 1732 kg/m$^3$ as has been defined previously for Bulk Bone [5, 13] and appropriate for homogeneous simulations.

## Validation

In order to test the validity of high frequency (low PPW) simulations ran here, a convergence test was run using the highest frequency sampled (1MHz), and thus the lowest PPW. In this test, CFL's of 0.1, 0.2, and 0.3 were used while PPW was varied between 3 and 6 by doubling dimensions of the simulation space for half of the simulations. All other parameters were held constant at 1 MHz Fundamental frequency, 12mm Bone thickness, 90 degrees trajectory (flat against bone), 80mm transducer depth and 80mm transducer width. The attenuation and refraction across this smaller parameter space are provided in the appendix (S1 Table); however, note that the simulation ran with the CFL and PPW values used in this study at 1Mhz (0.3 and ~2.96, respectively) varied from the theoretically most valid simulation (0.1 CFL and 6PPW) by a mere 5% in terms of attenuation (~0.68% vs. ~0.72%) and 5.56% in terms of refraction (17mm vs. 18mm). Note that this is through bone of the highest thickness and frequency most susceptible to error used in this dataset.

In addition, to further validate our simulations, a comparison between our 3D K-wave simulations and the classic O'Neil [14] solution was performed. To do so, we collected the lateral intensity (from the geometric center of the transducer) produced from each of our water simulations run at the highest Fundamental frequency (1MHz; 3PPW) and with a flat trajectory. These were compared to the lateral intensity projected by the O'Neil solution when provided with the same parameters (transducer diameter, radius of curvature, etc.). The overlap between the focal intensity predicted between these two methods is striking (see S1 Fig) and bolsters

the validity of our K-wave simulations. Indeed, the mean percent deviation of the area under the curve for the focal region (50% of maximum used as threshold) of the O'Neil and SMART_FUS-derived pressure vectors was 5.37%. An example image is present in the appendix (S1 Fig) while a folder with images for each parameter set tested in this way (42 in total) is included with our data at doi:10.5068/D1QD60.

## Discussion

SMART FUS enables simple and rapid estimation of skull attenuation and refractory effects when using LIFU in human subjects. This function is intended to aid in planning experiments and parameter selection. Furthermore, SMART FUS may be used to select parameters prior to performing higher-fidelity simulations. It provides an estimate of skull attenuation and refractory effects but should not be considered a replacement for high-resolution subject-specific simulations, which may provide higher validity for individual experimental settings and should be considered.

We selected methods that maximize validity while reducing individual simulation time to feasible levels to produce such a large dataset. Thus, some limitations exist. Several sacrifices regarding bone shape were made to reduce simulation time and avoid aliased skull models. The skull here was a simple flat, homogenous bone layer and did not approximate the heterogenous density and complex shape of real skull, which we found impossible to recreate given computational limitations. Previous work has demonstrated a difference in refraction between homogenous and inhomogeneous models of the same skull to vary between 0.6mm and 5.1mm while attenuation varied between 18.3% and 58.7% when aimed through the thick parietal bone [5]. This source of possible error should be considered when planning experiments using SMART_FUS and may further compel CT-derived skull models if available. However, the density and internal structure of subject skulls are, of course, impossible to know in the planning phase of an experiment where this toolbox finds much of its value.

While homogenous simulations are often used to reduce simulation complexity, skull-shape and reflection of energy off the back of the skull appear to dictate focal properties to some extent [5]. Furthermore, true skull models would not display homogenous density and material properties across the skull volume and can include areas of markedly lower density and pockets of air and fluid. The short simulation time and homogeneous bone layer chosen here do not allow us to account for these effects. However, the curved structures of more realistic skull models necessitate aliased simulation media, which are likely to reduce the validity of simulations considerably [11]. These were intentionally avoided here.

Similarly, complexity was reduced by assuming constant values for skull density and the intensity of emitted ultrasound. Skull density is known to differ substantially between individuals, which may dramatically impact the effect of skull on ultrasound propagation. We avoided a skull-density dimension to reduce simulation run-time and because skull-density of participants is usually unknown by experimenters unless computed tomography (CT) images are taken. In clinical settings where CT images are available, subject-specific simulations become much easier and are recommended. Future procedures may take advantage of increased computational power and add dimensions such as bone density or increase the resolution of the parameter space explored.

## Supporting information

**S1 Fig. O'Neil comparison example.** This figure shows the axial pressure (Pa) derived from SMART_FUS and O'Neil water simulations when using the same input parameters (in this case, a focal depth of 50mm and diameter of 60mm). Note that the x axis depicts the distance

from the back of the transducer's concavity and thus the focus is more than 50mm from this point. The noise between 0 and 15mm on the x axis in the SMART_FUS derived pressure map is also related to the fact that these values come from within the transducer concavity itself. (DOCX)

**S1 Table. Convergence test.** CFL's of 0.1, 0.2, and 0.3 were used while PPW was varied between 3 and 6 by doubling dimensions of the simulation space for half of the simulations. All other parameters were held constant at 1 MHz Fundamental frequency, 12mm Bone thickness, 90 degrees trajectory (flat against bone), 80mm transducer depth and 80mm transducer width. The attenuation and refraction across this smaller parameter space is provided. (DOCX)

## Acknowledgments

A thanks to Amir Vala Tavakoli, who helped write the readme.txt, reviewed the manuscript, and performed quality assurance on the Matlab files associated with this work.

## Author Contributions

**Conceptualization:** Joshua A. Cain.

**Data curation:** Joshua A. Cain.

**Formal analysis:** Joshua A. Cain.

**Investigation:** Joshua A. Cain.

**Methodology:** Joshua A. Cain, Shakthi Visagan.

**Project administration:** Martin M. Monti.

**Software:** Joshua A. Cain.

**Supervision:** Joshua A. Cain.

**Validation:** Joshua A. Cain.

**Visualization:** Joshua A. Cain.

**Writing – original draft:** Joshua A. Cain.

**Writing – review & editing:** Joshua A. Cain.

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
