## [Decision Letter · Decision Letter 0]

8 Apr 2022

PONE-D-22-03109S.M.A.R.T. F.U.S: Surrogate Model of Attenuation and Refraction in Transcranial Focused Ultrasound.PLOS ONE

Dear Dr. Cain,

Thank you for submitting your manuscript to PLOS ONE. After careful consideration, we feel that it has merit but does not fully meet PLOS ONE’s publication criteria as it currently stands. Therefore, we invite you to submit a revised version of the manuscript that addresses the points raised during the review process.

We look forward to receiving your revised manuscript.

Kind regards,

Talib Al-Ameri, Ph.D

Academic Editor

PLOS ONE

Journal Requirements:

Reviewers' comments:

Reviewer's Responses to Questions

**Comments to the Author**

1. Is the manuscript technically sound, and do the data support the conclusions?

Reviewer #1: Partly

Reviewer #2: Partly

Reviewer #3: No

2. Has the statistical analysis been performed appropriately and rigorously? 

Reviewer #1: I Don't Know

Reviewer #2: N/A

Reviewer #3: No

3. Have the authors made all data underlying the findings in their manuscript fully available?

Reviewer #1: Yes

Reviewer #2: No

Reviewer #3: No

4. Is the manuscript presented in an intelligible fashion and written in standard English?

Reviewer #1: Yes

Reviewer #2: Yes

Reviewer #3: No

5. Review Comments to the Author

Reviewer #1: The SMART FUS toolbox software that numerically models the acoustic propagation of low-intensity focused ultrasound (LIFU) is reported. The authors explained that users receive an estimation of the degree of refraction/attenuation expected for the given FUS parameters through rapid estimation of beam properties of LIFU transmitted through bone (skull).

The rationale behind the work is strongly justified and praiseworthy. The software is available to public, favoring dissemination of their work.

As authors indicated, the acoustic propagation through skull is complicated phenomena. Thus, any numerical model is desired to be supported by in vitro validation (not just modelling through models).

The following additional contents can strengthen their work;

1. Comparisons among existing simulation software and algorithms (although the work utilizes part of the 'k-wave' functions) in terms of performances indices.

2. Comparisons of simulation with respect to actual hydrophone mapping of the acoustic field, using quantifiable attenuators with different materials and shapes.

3. Considerations of modeling through inhomogeneous skull properties (although mentioned in the discussion).

4. As the work seems to be heavily dependent on the performance of k-wave algorithms, narrowing the scope of the report, for example, to a 'technical report', can be considered, if PlosOne supports such submissions.

5. Explanation of differential technical elements of their work from the k-wave and their separate performance validation.

Reviewer #2: The authors present a method for estimating the attenuation and aberration of focused ultrasound after propagating through a slab of bone. This uses a large dataset of pre-computed simulations, and interpolates between these values to give estimates of the attenuation and aberration given the transducer parameters and bone thickness. For people not familiar with running simulations, the idea is to give a quick way of estimating the effect of the bone.

Major comments:

1. There is no validation provided for the simulations. Particularly for the higher frequencies, I am not convinced that 3 points per wavelength is sufficient. Some validation must be provided. I would suggest at least the following:

(a) Transducer model in water: For a range of transducer sizes, angles, and frequencies (which relates to PPW in this study), verify that the axial pressure in water agrees with a suitable reference solution (e.g., computed using the O'Neil solution, the Rayleigh integral, or the FOCUS toolbox, etc).

(b) Propagation through bone: A convergence test should be performed to show that the results don't depend on the sampling parameters. In other words, run a subset of simulations (particularly for the highest frequency) at higher points per wavelength and smaller CFL, and show the predicted field doesn't change.

(c) Do the predictions match the data presented in https://doi.org/10.48550/arXiv.2202.04552? It should be straightforward to compare the precomputed field from BM3-SC1 (single layer bone) with BM1-SC1 (water), and compare them with the values returned by SMART_FUS. I intended to do this, but couldn't run the SMART_FUS software (see comment below).

2. No evidence is given that a single layer flat bone is a good surrogate for estimating the attenuation / aberration through a real skull (e.g., which has non-uniform thickness and interior structure). Some justification / evidence should be provided.

Minor comments:

3. If the simulations only take 10s, is there an argument for end users just running the simulations themselves for their specific parameters?

4. How is the distance between the transducer and the bone surface determined? Is it fixed?

5. What about the attenuation within the skin and brain?

6. I was unable to run the SMART_FUS functions from the Zenodo link as the data files .\\data\\5d_mats\\attentuation_matrix.mat and .\\data\\5d_mats\\refraction_matrix.mat are missing.

7. Do you have permission to redistribute the imdistline function from the MATLAB image processing toolbox in an open repository?

8. In Fig 1B, should the right plot title be refraction?

9. DS appears in the author contributions but is not in the author list.

Reviewer #3: This is a short manuscript that introduces two matlab functions to aid in transcranial focused ultrasound. The functions are idealized functions of a single bone type of a flat configuration. As such a simplified model of bone, it is not clear what the application for these matlab functions would be. A better understanding of the intended use of the functions is needed here. A clear use case would be helpful.

In addition, some details of the functions are not clear. What is “trajectory?” Is this angle with respect to the bone surface?

What was the rationale and references for the choices of bone acoustic properties?

6. PLOS authors have the option to publish the peer review history of their article (what does this mean?). If published, this will include your full peer review and any attached files.

Reviewer #1: No

Reviewer #2: No

Reviewer #3: No

---

## [Author Response · Author response to Decision Letter 0]

2 Jun 2022

All requested changes have been made or addressed in the attached files.

---

## [Decision Letter · Decision Letter 1]

29 Jul 2022

PONE-D-22-03109R1S.M.A.R.T. F.U.S: Surrogate Model of Attenuation and Refraction in Transcranial Focused Ultrasound.PLOS ONE

Dear Dr. Cain,

Thank you for submitting your manuscript to PLOS ONE. After careful consideration, we feel that it has merit but does not fully meet PLOS ONE’s publication criteria as it currently stands. Therefore, we invite you to submit a revised version of the manuscript that addresses the points raised during the review process.

We look forward to receiving your revised manuscript.

Kind regards,

Talib Al-Ameri, Ph.D

Academic Editor

PLOS ONE

Reviewers' comments:

Reviewer's Responses to Questions

**Comments to the Author**

1. If the authors have adequately addressed your comments raised in a previous round of review and you feel that this manuscript is now acceptable for publication, you may indicate that here to bypass the “Comments to the Author” section, enter your conflict of interest statement in the “Confidential to Editor” section, and submit your "Accept" recommendation.

Reviewer #2: (No Response)

2. Is the manuscript technically sound, and do the data support the conclusions?

Reviewer #2: Partly

3. Has the statistical analysis been performed appropriately and rigorously? 

Reviewer #2: N/A

4. Have the authors made all data underlying the findings in their manuscript fully available?

Reviewer #2: Yes

5. Is the manuscript presented in an intelligible fashion and written in standard English?

Reviewer #2: Yes

6. Review Comments to the Author

Reviewer #2: Accuracy: I thank the authors for their revision. However, my fundamental concern from the previous version - are the simulations correct? - has still not been adequately addressed. While k-Wave as a solver may have previously been validated under certain conditions, it is essential that you validate its use under the conditions in which you use it for this study. As a minimum, I would like to see a comparison of the axial pressure from the transducer at the lowest sampling used (3 points per wavelength) against an analytical solution (e.g., the O'Neil solution). This is easy to do, as the O'Neil solution is also implemented in k-Wave.

Convergence test: The convergence tests mentions data in an appendix, but I couldn't see an appendix?

Flat skull bone: There is still no evidence provided that the use of a flat skull bone is a good surrogate for a real skull which varies in thickness. I appreciate that there might be some computational difficulties as discussed, and I'm happy with the homogeneous vs heterogeneous justification, but flat vs real is still not justified. If you took a real skull, and a flat skull with the same nominal thickness, what errors might be introduced? I could imagine in some scenarios they might be be considerable. Quantifying the errors is a important concern if the authors intend the toolbox to be used by other (non-computational) groups.

Attenuation in the brain: While the density and sound speed are similar, the acoustic attenuation in soft tissue is 2 orders of magnitude higher, so the question remains.

Do the predictions match the data presented in ...: This was intended as a suggestion to validate the results of the paper, not as a homework problem for the reviewers.

7. PLOS authors have the option to publish the peer review history of their article (what does this mean?). If published, this will include your full peer review and any attached files.

Reviewer #2: No

---

## [Author Response · Author response to Decision Letter 1]

29 Aug 2022

Reviewer #2: Accuracy: I thank the authors for their revision. However, my fundamental concern from the previous version - are the simulations correct? - has still not been adequately addressed. While k-Wave as a solver may have previously been validated under certain conditions, it is essential that you validate its use under the conditions in which you use it for this study. As a minimum, I would like to see a comparison of the axial pressure from the transducer at the lowest sampling used (3 points per wavelength) against an analytical solution (e.g., the O'Neil solution). This is easy to do, as the O'Neil solution is also implemented in k-Wave.

We have performed the requested comparison to the O’Neil solution (O'Neil, H. Theory of focusing radiators. J. Acoust. Soc. Am., 21(5), 516-526, 1949) for each parameter set at the 3 PPW and SMART_FUS produces very similar pressure maps to the O’Neil solution. The procedure and its results are described in the main body of text in a new “Validation” section (line 175). However, note that the integral of the focal region (everything over 50% of maximum pressure) for the O’Neil solution-derived and SMART_FUS-derived pressure maps differed by 5.56% across 42 different parameter sets. An example comparison image is provided here, below. Note that no new k-Wave simulations were run but this is a comparison between the provided dataset and a newly generated pressure field created using the O’Neil solution. 

Convergence test: The convergence tests mentions data in an appendix, but I couldn't see an appendix?

We apologize for the fact that the appendix was somehow not included in the response you received. We have made sure to include it this time and will follow up with the editor to make sure you receive it. We believe that the convergence test data, in addition to the new O’Neil comparison, will satisfy your concerns. 

Flat skull bone: There is still no evidence provided that the use of a flat skull bone is a good surrogate for a real skull which varies in thickness. I appreciate that there might be some computational difficulties as discussed, and I'm happy with the homogeneous vs heterogeneous justification, but flat vs real is still not justified. If you took a real skull, and a flat skull with the same nominal thickness, what errors might be introduced? I could imagine in some scenarios they might be be considerable. Quantifying the errors is a important concern if the authors intend the toolbox to be used by other (non-computational) groups.

We apologize for not responding to the flat bone concern in detail in the reviewer 2 response. We had done so in more detail to reviewer 3. The two simple reasons why curved bone was avoided are that 1) greatly increased spatial resolution is necessary to emulate curved bone in order to eliminate the discrete grid’s staircase effect at the material interfaces and 2) since the purpose of this package is to aid in the planning of experiments, it is unclear what particular shape the reviewer would have us select for our simulations. A convexly curved skull cannot be assumed because many targets may require passage through the relatively flat or even concavely curved side of the skull.

1) Again, a flat bone structure avoids the staircase effect that arises in any discrete grid simulations studying more complex skull models, which was found to introduce the most error among possible error-inducing factors for similar simulations using k-Wave (Robertson et al., 2017; cited in the main text). I would further argue that one might expect more error to be introduced by assuming a particular curvature in our simulations, for which the purpose is to provide estimates that are relevant to a broad range of skull shapes. One could argue that a range of curvatures may be assessed as another variable or tunable parameter in our simulations. However, due to the complexity required, this could necessitate a higher-dimensional simulation space, which would multiply the time required to create this dataset greatly. At best, a new dimension must be added to the simulations (skull curvature), which would multiply the time required to create this dataset by as many curvatures as are deemed required. Inevitably, computer simulations require some sacrifices to complexity/validity. Given that the increased spatial resolution required to emulate curvatures at material interfaces on our discrete grid simulations in order to reduce staircasing artifacts is itself a major source of computational complexity and runtime, we believe that opting to not create simulations with bone curvature is justified or even highly preferable. Note that staircasing has been shown to introduce more error than e.g., “the influence of the BLI, changes in the effectiveness of the absorbing perfectly matched layer (PML), the impact of numerical dispersion, the representation of discontinuities in medium properties (Robertson et al., 2017)”

2) Given great individual differences in skull shape which may be present directly under the transducer when positioned for any given stimulation session with any given individual, it seems impossible to better estimate that shape prior to subject recruitment. Instead, we opt here to simply represent bone at a given width, which we believe can be more readily estimated for a given planned trajectory (e.g., into the central thalamus through the temporal bone) using the known ranges of human skull thickness and the known position of brain regions of interest. For instance, say a slightly thicker bone medium were present on one side of the bone just under the transducer. How ultrasound interacts with this structure would depend on if the transducer were angled towards or against this structure. Accounting for such a possibility would necessarily introduce several new dimensions to our simulation space without getting close to accounting for all the complexity possible in real-world scenarios. We believe this package retains its value despite not representing the very many ways human skull shapes can differ, especially considering that the staircasing necessary to emulate skull curvature itself is likely to introduce considerable error.

Attenuation in the brain: While the density and sound speed are similar, the acoustic attenuation in soft tissue is 2 orders of magnitude higher, so the question remains.

When models that included different properties for brain (a detailed layered cortical model) were compared to those that assumed brain possessed the same properties as water, only insignificant changes in pressure maps were observed (see https://iopscience.iop.org/article/10.1088/1741-2560/13/5/056002). This data has been used to justify the absence of brain models in more recent works (https://iopscience.iop.org/article/10.1088/1741-2552/aa843e). 

However, it should also be noted that all numerical measures provided by SMART_FUS are relative between simulations that include bone and those that do not. Thus, the attenuation caused by soft tissue on one side of the bone layer would be identical between simulations that include bone and those that do not. Thus, the contribution of bone to overall attenuation/refraction provided by SMART_FUS is likely to remain highly valid even when ignoring the differences in acoustic properties in water and soft tissue. In other words, the relative attenuation/refraction expected between a BONE+WATER+BRAIN and a WATER+BRAIN model is likely to be very similar to that expected between a WATER simulation since the only difference in both cases is the presence of BONE. Complex interactions between each medium that may invalidate this logic have been empirically shown to be negligible (https://iopscience.iop.org/article/10.1088/1741-2560/13/5/056002) and anyhow are likely to be even less relevant to the simple simulations used to create SMART_FUS. 

Do the predictions match the data presented in ...: This was intended as a suggestion to validate the results of the paper, not as a homework problem for the reviewers.

We apologize. I believe we misunderstood the question. Now we understand that you wanted us to quantitatively compare the bone and water simulations performed in Aubry et al., 2022 by downloading and analyzing their data. We have now done this. We found that, when comparing simulations “PH1-BM1-SC1_KWAVE.mat” and “PH1-BM3-SC1_KWAVE.mat”, which differ only in the presence of bone, there is an attenuation of 61.7% and a refraction of 6.5mm produced by the 6.5mm bone layer. When running SMART_FUS.m using the same parameters (Flat 90-degree Trajectory, 500 kHz AF, 64mm diameter, 64mm focal distance, 6.5mm bone) it estimates that 56.8% attenuation and a 9.18mm refraction. It is important to note that the simulations run in Aubry et al., 2022 are not identical to SMART_FUS.m in that the transducer is moved away from the bone by 30mm (which would be uncommon in experimental settings), compared to the 0mm chosen for SMART_FUS.m simulations, which we believe explains the arguably relevant ~2.5mm difference in expected refraction between both estimates. In general, we feel that the similarity between these estimates as well as the qualitative similarity between the energy fields of Aubry et al., 2022’s high-validity simulation (shown below; Aubry left, SMART_FUS, right) bolster the validity of SMART_FUS.m.

---

## [Editor Report · Decision Letter 2]

19 Sep 2022

S.M.A.R.T. F.U.S: Surrogate Model of Attenuation and Refraction in Transcranial Focused Ultrasound.

PONE-D-22-03109R2

Dear Dr. Cain,

We’re pleased to inform you that your manuscript has been judged scientifically suitable for publication and will be formally accepted for publication once it meets all outstanding technical requirements.

Kind regards,

Talib Al-Ameri, Ph.D

Academic Editor

PLOS ONE
---

## [Editor Report · Acceptance letter]

20 Oct 2022

PONE-D-22-03109R2 

S.M.A.R.T. F.U.S: Surrogate Model of Attenuation and Refraction in Transcranial Focused Ultrasound. 

Dear Dr. Cain:

I'm pleased to inform you that your manuscript has been deemed suitable for publication in PLOS ONE. Congratulations! Your manuscript is now with our production department. 

Kind regards, 

on behalf of

Dr. Talib Al-Ameri 

Academic Editor

PLOS ONE